# Using Radiomics-Based Machine Learning to Create Targeted Test Sets to Improve Specific Mammography Reader Cohort Performance: A Feasibility Study

**DOI:** 10.3390/jpm13060888

**Published:** 2023-05-24

**Authors:** Xuetong Tao, Ziba Gandomkar, Tong Li, Patrick C. Brennan, Warren Reed

**Affiliations:** 1Discipline of Medical Imaging Science, Faculty of Health Sciences, The University of Sydney, Sydney, NSW 2006, Australia; 2The Daffodil Centre, The University of Sydney, Sydney, NSW 2006, Australia; 3Sydney School of Public Health, Faculty of Medicine and Health, The University of Sydney, Sydney, NSW 2006, Australia

**Keywords:** mammography, mammography interpretation, diagnostic errors, radiomics, machine learning

## Abstract

Mammography interpretation is challenging with high error rates. This study aims to reduce the errors in mammography reading by mapping diagnostic errors against global mammographic characteristics using a radiomics-based machine learning approach. A total of 36 radiologists from cohort A (*n* = 20) and cohort B (*n* = 16) read 60 high-density mammographic cases. Radiomic features were extracted from three regions of interest (ROIs), and random forest models were trained to predict diagnostic errors for each cohort. Performance was evaluated using sensitivity, specificity, accuracy, and AUC. The impact of ROI placement and normalization on prediction was investigated. Our approach successfully predicted both the false positive and false negative errors of both cohorts but did not consistently predict location errors. The errors produced by radiologists from cohort B were less predictable compared to those in cohort A. The performance of the models did not show significant improvement after feature normalization, despite the mammograms being produced by different vendors. Our novel radiomics-based machine learning pipeline focusing on global radiomic features could predict false positive and false negative errors. The proposed method can be used to develop group-tailored mammographic educational strategies to help improve future mammography reader performance.

## 1. Introduction

Self-assessment mammography test sets have been demonstrated to be an effective learning approach in improving the clinical performance of mammography readers, where participants are asked to assess a mammography set enriched with cancer cases, displaying a range of image appearances with known truth [1,2]. Upon completion, participants receive immediate, individual feedback on their performance, including sensitivity and specificity, allowing them to anonymously compare their results with their peers and learn from their mistakes by visualizing their errors (Figure 1). This exposure to a wide range of mammographic appearances, the high prevalence of cancer cases, and the immediate feedback with visual display make self-assessment mammography test sets an effective approach for developing radiologists’ expertise [3]. However, most self-assessment modules available use a one-size-fits-all approach to selecting training cases without considering the particular needs of specific cohorts of radiologists [4]. An alternative solution for customizing the test sets used would be cohort-tailored self-assessment test sets, with an emphasis on mammographic appearances that are particularly challenging for that group [5]. This is especially important in continuing education programs, where participants may have varying levels of experience and domain knowledge, and the performance data for each individual is unavailable, so personalizing the test set is not possible. In this instance, customizing the educational materials according to the readers’ experience and expertise level as a cohort is highly desirable as an ideal education strategy that focuses on improving their weaknesses to enhance learning outcomes. Artificial intelligence (AI) can quantify image analysis and recognize complex patterns, making it a potential solution for curating such tailored self-assessment cohort test sets [6].

The application of AI, often in the form of machine learning, is transforming education by improving instructor–learner interactions and increasing student access to educational materials [7,8]. However, customizing educational content based on the specific needs of each cohort of learners in medical imaging has not yet been fully developed [9,10]. Previous studies have explored the potential of AI in radiology education and investigated the relationship between image characteristics and radiologists’ diagnostic errors, but the results have been limited and inconsistent. For example, one study mapped the errors of radiology trainees in distinguishing malignant from benign breast masses to two features (breast density and mass margin) from the Breast Imaging-Reporting and Data System (BI-RADS) using AI [11]. This study found that machine learning models can predict individual trainees’ diagnostic errors using two BI-RADS features [11]. However, the number of readers involved in the study was small, and they only investigated segmented breast masses instead of the entire mammographic case [11]. Another study explored the relationship between wavelet-decomposed mammographic features and radiologists’ diagnostic errors and found that artificial neural networks, another form of AI, can predict radiologists’ decision outcomes on the eye-attracted local areas [12]. However, the prediction accuracy was inconsistent, and more features needed to be investigated [12]. To improve the prediction performance, radiomics, an emerging technique in image analysis, should be explored.

Radiomics, a process of high-throughput extraction of image features into minable quantitative data, has been proven to help predict radiologists’ diagnostic errors [13]. In this study, we aimed to test the hypothesis that a radiomics-based machine-learning approach can help identify challenging mammographic cases for a specific reader cohort and thereby improve the curation of cohort-tailored test sets. To test the hypothesis, we developed machine-learning models to map the global mammographic characteristics, represented by a comprehensive set of image-derived radiomic features, to the actual difficulty level of the mammographic case for two independent cohorts of mammography readers. The readers in each cohort shared similar demographic characteristics. One cohort was from an Asian country with both mass and opportunistic mammographic screening practices but with a low participation rate (cohort A); the other was from a Pacific country with a nationwide population-based mammography screening program and a high screening rate (cohort B). To our knowledge, this is the first study to investigate cohort-specific error-making patterns using a comprehensive and systematic set of radiomic features. The proposed method, if successful, could help us to further understand the cohort-based weaknesses in reading mammograms and develop group-tailored educational strategies to improve future mammography reader performance.

## 2. Materials and Methods

### 2.1. Mammogram Acquisition

This study analyzed 60 high-density mammographic cases collected from breast screening programs. Each case was deidentified and contained standard bilateral craniocaudal (CC) and mediolateral oblique (MLO) mammograms. Of the 60 cases, 40 were cancer-free with no benign findings, evaluated by at least two senior radiologists, followed by two-year negative screening reports, while the remaining 20 cancer cases (lesions = 21) were biopsy-proven. The locations of the cancers were provided by a senior radiologist, who did not participate in the subsequent study, based on mammograms, histopathological reports, and any additional images. Information on breast density, lesion types, and manufacturers is summarized in Table 1.

### 2.2. Mammography Case Difficulty Analysis

Two separate cohorts of radiologists (cohort A, *n* = 20 and cohort B, *n* = 16) read all mammographic cases in a standardized mammography reading environment. All the participating readers were certified radiologists. The mammograms were displayed in DICOM format, and all typical post-processing (e.g., zooming, panning, windowing) options were available. Each reader independently examined all mammograms, marked all suspicious lesions with a mouse-controlled cursor, and reported radiological findings using the Royal Australian and New Zealand College of Radiologists (RANZCR) rating system (1- normal, 2- benign, 3- indeterminate, 4- suspicious of malignancy, and 5- definitely malignant). Although the RANZCR and BI-RADS grading systems demonstrate some differences, the two systems are transferable, with a RANZCR rating of 3 being equivalent to BI-RADS 3 and 4A and the RANZCR rating of 4 being equal to BI-RADS 4B and 4C. Ratings 1, 2, and 5 were interchangeable in the two systems [14]. Each reader was allowed to annotate multiple areas; however, only the one with the highest rating was preserved for analysis. All readers were unaware of the cancer prevalence and lesion types in this image set prior to reading and blinded to any clinical information and additional imaging. However, they were told that the cancer prevalence in this test set was higher than the incidence rate in the screening setting. More details about this procedure have previously been described [15].

We calculated the group-specific case difficulty for each mammographic case based on radiological assessment and cancer truth data. Specifically, the RANZCR rating of 1 or 2 was considered negative, while 3 and above were positive. The location of a cancer lesion was considered correct if the annotated point was 250 pixels within the actual cancer location on DICOM images. The case difficulty level was measured as the proportion of radiologists in each group making diagnostic errors in the form of false positives (for cancer-free cases), false negatives (for cancer cases), and cancer location errors (for cancer cases). Specifically for cancer cases, the difficulty level calculated based on the proportion of false negatives was called the case-based difficulty, while the one calculated based on the proportion of location errors was the lesion-based difficulty. Based on the proportion of false positives, the cancer-free cases were divided into three approximately equal groups, ‘easy’, ‘median’, and ‘difficult’, indicating different levels of difficulty. Similarly, the cancer cases were grouped into the same categories of difficulty, with roughly equal numbers in each category based on false negatives (case-based) and lesion location errors (lesion-based).

### 2.3. Image Pre-Processing

The breast region was segmented from the background, and in the MLO views, the pectoral muscle area was removed using a previously validated pipeline based on thresholding, morphological transformation, and hough line detection algorithms [16]. Then, three ROIs were delineated for subsequent feature extraction, including the largest square of the central breast region (square), the retroareolar area (RA), and the whole breast region (whole) (Figure 2). This was done using the open framework for mammography image analysis in Matlab_R2022b (Mathworks, Natick, MA, USA) [17,18,19]. The central and retroareolar regions of the breast were investigated because these areas contained the most useful textural information of fibrogranular tissue, which has been related to the risks of developing breast cancer [20,21]. The entire breast region was also investigated, as neither the central nor the retroareolar area can fully reflect the heterogeneity of the breast tissue [22]. Subsequently, z-score normalization was applied to each ROI to minimize the grey-scale variations across image acquisitions. The original ROIs before normalization were also preserved to evaluate the impact of normalization on model performance.

### 2.4. Extraction of Radiomic Features

A total of 203 radiomic features were extracted from each normalized and original ROI. These features have been studied extensively in previous studies for their application in breast cancer diagnosis and risk assessment. They can be grouped into three broad categories: histogram-based first-order statistics, texture-based second-order features, and transform-based higher-order features [23]. The histogram-derived features describe the distribution of grey-level intensities; the textural descriptors quantify the spatial relationship between pixels; and the transform-based features, extracted from the decomposed images using a bank of filters, provide additional information on the interrelationship among pixels [24]. Specifically, there were 28 histogram-based statistics and 159 textual features, including 88 from the grey level co-occurrence matrix (GLCM) [25], seven from the grey level run length matrix (GLRLM) [26], six from the grey level sharpness measure (GLSM) [27], 15 grey level difference statistics (GLDS) [28], 15 from the neighborhood grey tone difference matrix (NGTDM) [29], eight from the statistical feature matrix (SFM) [30], 18 laws texture energy measures [31] and two from fractional dimension texture analysis [32]. Additionally, there were 16 transform-based features, including six and eight from Gabor- and RFS- filtered images [33,34], respectively, and two features from the Fourier-transformed images [32]. The radiomic features used in this study can be found in Appendix A.

### 2.5. Feature Selection and Model Construction

As shown in Figure 3, we developed three machine-learning pipelines to predict case difficulty levels for both cancer-free and cancer cases. Specifically for cancer cases, models were developed to predict case-based and lesion-based difficulty levels separately. Before implementation, mammography cases from the median difficulty group were excluded, as including two more extreme categories of difficulty accentuates differences between groups and allows a good separation between the most ‘difficult’ and ‘easy’ cases for group readers. Of note, 13 and 10 out of 40 cancer-free cases were excluded for cohort A and cohort B, respectively. Five and four out of 20 cancer cases were excluded from the pipeline predicting case-based difficulty levels, and six and five out of 20 cancer cases were excluded from predicting lesion-based difficulty levels for cohorts A and B, respectively. 

Within each pipeline, the random forest (RF) classifier was chosen as the learning model since it is a non-linear algorithm consisting of an ensemble of decision tree classifiers. Although each tree tends to suffer from instability and overfitting, RF aggregates the results from all trees and makes a relatively more accurate prediction with better generality. Specifically, for cancer-free cases, four RF models were trained on features from CC and MLO views of both breasts, respectively, as false positive findings can occur on either side of the breasts. However, for cancer cases, only the features from the lesion-located views were fed into the pipeline for training, which is because, on the existence and location of cancer on the lesion side, the appearance of the contralateral normal breast offers little information if there is any, and therefore cannot help in predicting the proportion of readers who missed or mislocated the lesion. As shown in Figure 3, each pipeline returned multiple class probabilities indicating the likelihood of a given view being ‘difficult’. However, to obtain a case-wise prediction, we fused all outputs using the ‘maximum probability’ approach, and only the one with the highest probability score was considered as the final output of the pipeline. This method was taken to ensure the pipeline’s sensitivity to detect challenging mammographic cases for educational purposes. Ideally, a case should be labeled as ‘difficult’ even if only one or two views present actual detection difficulty, regardless of the error status in the remaining views.

To avoid data leakage and overfitting during training, we applied nested cross-validation for feature selection and model construction (Figure 4). The outer layer used leave-one-out cross-validation to estimate the classifier’s generability on unseen data; the inner loop adopted repeated three-fold cross-validation with three repeats to train the model, select features and simultaneously adjust the hyperparameters in RF. To select features, we applied RF embedded method because it required less computation than the wrapper approach and was more robust than the filter approach [35]. In addition, it allowed simultaneous feature selection and model training so that the optimal number of features to be selected could be considered as a hyperparameter of the pipeline, which can be tuned together with the number of trees (from 50, 100, and 500 to 1000), the maximum sample size (from 10%, and 50% to 100%), and the depth of each tree (from 1 and 5 to 10). These hyperparameters were independent of the input dataset but determined how well the RF model could be trained to map the input features to the output difficulty level. We used the exhaustive grid search method to select the best combination of hyperparameters that produced the highest averaged AUC value in the inner validation set (Figure 4b). The combination of hyperparameters with the best validation performance was further evaluated on the outer testing set in the outer layer. Model performance was measured by sensitivity, specificity, accuracy, and AUC, with AUC equal to 0.5 considered random guessing. The feature selection and model-building processes were implemented in the Scikit-learn package in Python Version 3.9.4.

Additionally, a 2000-time stratified bootstrap was used to obtain the 95% confidence interval of AUC for model evaluation. Additionally, we compared the AUC values between cohort A and cohort B using a 2000-time stratified bootstrap method. Additionally, we investigated the effects of ROI placement and feature normalization on error detection using DeLong’s test. The two-tailed tests of significance were applied using a significance level of 0.05. The statistical analyses were performed in the ‘pROC’ package in R Version 4.2.2.

## 3. Results

Table 2 summarizes the performance of models on the testing dataset in error detection using normalized and original ROIs for readers from cohort A and cohort B, respectively. As these figures suggested, the machine learning model using global radiomic features could detect false positive and false negative errors but was not consistently successful in detecting location errors for both cohorts. For cohort A, the average AUCs of models detecting false positives, false negatives, and location errors were 0.899, 0.660, and 0.447; and for cohort B, the average AUCs in detecting false positives, false negatives, and location errors were 0.754, 0.729, and 0.598.

Comparisons of AUCs between cohort A and cohort B using a 2000-time stratified bootstrap indicated that the model built on features of normalized square-predicting false positives (cohort A: 0.882; cohort B: 0.651) and the model built on features of non-normalized whole-predicting false positives (cohort A: 0.933; cohort B: 0.702) showed significant differences with higher predictive values observed for cohort A. *p*-values are shown in Table 3. We also investigated the impacts of ROI placement and feature normalization on error detection, and DeLong’s test revealed that these factors had statistically insignificant effects with *p*-values greater than 0.05 (Table 4 and Table 5).

## 4. Discussion

This study was designed as a proof-of-concept study, exploring the feasibility of using AI to produce customized self-assessment educational test sets containing difficult-to-interpret mammographic images to improve the performance of a certain group of mammography readers. Here, we considered two cohorts of radiologists and investigated how mammographic characteristics relate to the diagnostic decisions of mammography readers using machine learning methods and radiomics techniques. Mapping image features to interpretative errors can help design mammography training content adapted to the specific needs of the readers. To achieve this, we proposed a generalized model to automatically select mammography cases from a large image dataset that can match the weaknesses and strengths of a group of mammography readers with similar characteristics for tailored training. Unlike a user-based model that identifies individual error-making patterns, our model can predict common pitfalls using the difficult cases associated with a high probability of diagnostic errors for most readers in a group by analyzing the previous readings of the cohort of readers. Although prediction of the collective model might sacrifice a certain degree of precision on the individual level, it is more time-efficient and achievable since individual reading data are often inadequate for model construction using data-driven machine learning methods [36]. Additionally, the individualized model might not always reflect the true reading weaknesses of each reader as the retrospective analysis of previous reading data might be biased if other factors, such as poor image quality or fatigue, affect the actual performance of the specific reader [37,38]. However, such bias can be counteracted using this collective cohort-based method, producing a more stable prediction of the generically challenging cases for a specific group.

This study also showed that radiologists from the same country share certain error-making patterns when reading mammography, regardless of the variation in the reading experience, which makes country-specific training in mammography reading a feasible solution to improve diagnostic performance. This finding was consistent with previous research showing variability in the type of breast lesions most likely to be missed among radiologists across countries [39]. For example, one study found that spiculated stellate lesions were more likely to be neglected by Vietnamese radiologists (31); however, non-specific density was difficult to detect by Australian radiologists [40]. The reasons for these variations are multifaceted and may be due to different focuses in mammographic training in different countries and/or radiologists’ various experience with certain types of mammographic densities or features [15].

Furthermore, we verified the value of global radiomic features for predicting false positive and false negative errors, while the lesion location error was less predictable, which might require more localized features. This may suggest that the diagnostic decision on the case or image-based normality or abnormality was related to the overall parenchymal pattern of breast tissue; however, the whole appearance of the breast provided inadequate information to affect the decision on cancer location. This may be because the location signal was overwhelmed by the relatively strong global signal, which contained much irrelevant background information. This supports the findings of Brennan et al. and Evans et al., who found that radiologists showed above-chance performance in distinguishing abnormal from normal mammograms in the first impression of the image, but the signal from this quick glimpse did not contain information on the location of suspicious areas [41,42]. Recently, it was also shown that combining the global image characteristics with a deep learning-based computer-aided detection tool for mammography can improve its performance in cancer detection [14,43].

Although radiomic models showed desirable performance in detecting false positives and false negatives for both cohorts, radiologists in cohort B (the country with a population-based breast screening program) were less predictable compared to those in cohort A (an Asian country with both mass and opportunistic mammographic screening practices with low participation rate). This may be related to study readers from cohort B outperforming those from cohort A and therefore making fewer diagnostic errors. With a limited number of errors, it is likely that detecting error-making patterns becomes more difficult for machine-learning approaches. This has implications around the value of a radiomic pipeline among high-performing radiologist cohorts whose diagnostic errors tend to be less predictable. In other words, a pipeline may be more helpful in identifying and eliminating errors made by less experienced radiologists in mammography reading, such as radiologist trainees, medical students, and general radiologists not specializing in mammography. However, once these radiologists gain expert diagnostic skills, these errors may not be easily detected, and a more individualized prediction model would be more likely to be helpful [14].

Additionally, we found that feature normalization provided no beneficial effect on the prediction of error patterns, although these mammograms were captured by machines from different manufacturers. This differs from other radiomic studies, such as prognostic or cancer subtype classification research, which predicts the probability of an outcome at the molecular level. In clinics, this is usually achieved through pathological tests, and although mammograms can contain biological and genetic information related to cancer progression and subtyping, it is often hidden from the radiologist [44]. Therefore, standardization of radiomic features may provide additional benefits in these studies to unravel image characteristics that are not immediately apparent to the vision of the radiologist [45]. However, in image perception studies where radiologists interpret the original rather than any transformed radiographs, it makes sense that the non-normalized features can reflect the clinical image processing procedure of radiologists and can perform well in diagnostic error prediction.

As a proof-of-concept study, it has several limitations. First, the sample size was small, which could potentially impact the generability of the results for larger populations. External validation using publicly available datasets would be valuable for model validation; however, for a study focussing on predicting reader diagnostic errors, no current publicly available datasets containing mammographic examinations were suitable, as these datasets are primarily designed to develop AI models for diagnosis, while our objective is to predict the likelihood of diagnostic errors made by radiologists. Despite the sample size limitation, we collected radiological assessments from 36 radiologists with varying levels of experience, and the mammographic dataset is enriched with cancer cases representing various lesion subtypes commonly encountered in breast screening. This ensures representative observer performance, which is crucial in diagnostic error analysis studies. Second, no localized features were used, possibly explaining why the location errors of the lesion were unpredictable. Nonetheless and despite these limitations, this study highlights the importance of global radiomic features in the prediction of cohort-specific diagnostic errors. Therefore, we encourage researchers to explore these features, alongside localized features, to uncover their potential in various tasks such as diagnostic error prediction, breast cancer detection, and breast cancer prediction. We also hope that our study stimulates reflection on the current one-size-fits-all radiology training system, encouraging a collective effort to revolutionize radiological education. We encourage researchers to embrace the novel ideas proposed in this study, leveraging the potential of AI to improve diagnostic performance, and we urge the community to create more comprehensive and extensive reader performance datasets.

## 5. Conclusions

We proposed a framework to detect false positive and false negative errors for cohort-based radiologists in reading high-density mammograms using global radiomic features. This method can be further validated, optimized, and then integrated into computer-assisted adaptive mammography training for future group-tailored educational strategies to help inform greater error reduction in mammography.

## Figures and Tables

**Figure 1 jpm-13-00888-f001:**
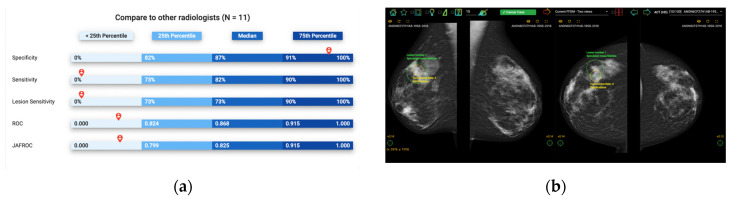
Illustration of Breastscreen REader Assessment STrategy (BREAST) platform for reading mammographic images. (**a**) Once readers completed the assessment, they were ranked against their peers anonymously. (**b**) A visual display of all errors and the truth.

**Figure 2 jpm-13-00888-f002:**
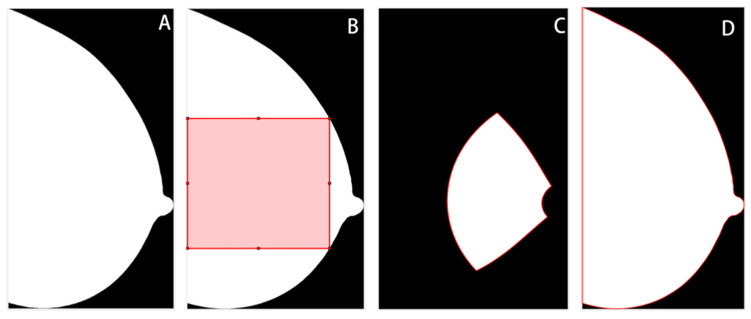
Illustration of regions of interest (ROIs). (**A**) Whole breast without segmentation. (**B**) The largest square from the central breast region. (**C**) Retroareolar region. (**D**) Whole breast region.

**Figure 3 jpm-13-00888-f003:**
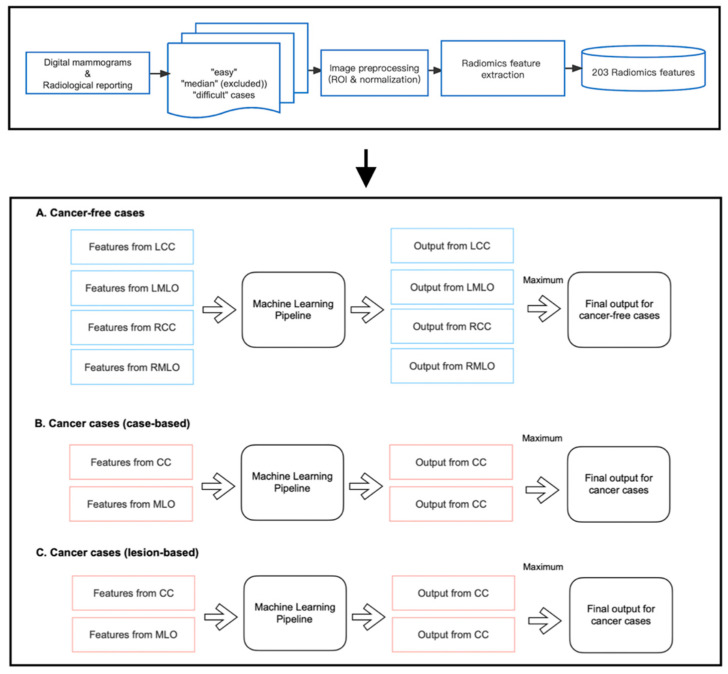
Implementation of radiomic models.

**Figure 4 jpm-13-00888-f004:**
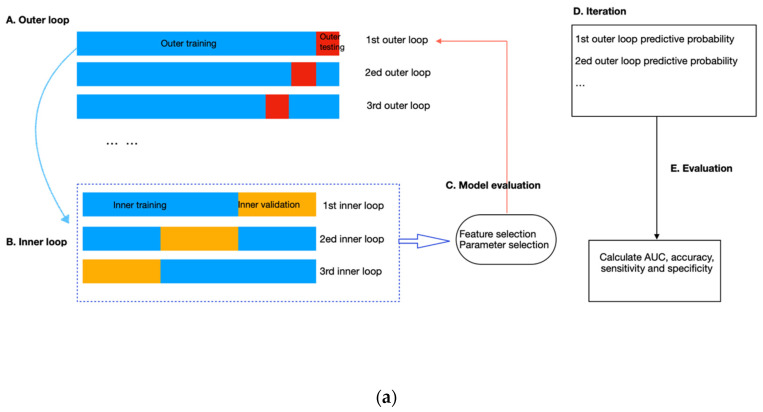
(**a**) Flow diagram of the nested cross-validation. (A) The outer loop of the nested CV was divided into k folds (k = the number of samples), of which the (k-1) folds were treated as the outer training set and the kth fold as the outer testing set. (B) Data from the outer training set flowed into the inner loop, being further split into three folds, two of which were the inner training set, and the remaining fold was the inner validation set. Within the inner layer, the number of features and trees, the maximum sample size, and the depth of each tree were selected using an exhaustive grid search method. (C) The RF classifier with the highest AUC scores in the inner loop validation set was evaluated in the outer testing, giving the corresponding predictive probability. (D) The workflow (A–C) was repeated until all samples in the outer loop were tested. (E) In the end, an AUC score incorporating the results of all outer loops was calculated as the classification performance of the predictive model. (**b**). Schematic of feature selection and hyperparameter tuning within the inner loop. All combinations of hyperparameters will be exhaustively investigated using repeated three-fold cross-validation, and the performance on the inner validation set from the three-folds will be averaged. The hyperparameter combination with the highest averaged performance, represented as the AUC score, will be further evaluated on the outer testing set in the outer loop.

**Table 1 jpm-13-00888-t001:** Details of mammography cases.

Factors	Non-Cancer Cases (*n* = 40)	Cancer Cases (*n* = 20)
Breast density		
BI-RADS A	0	0
BI-RADS B	2	0
BI-RADS C	30	15
BI-RADS D	8	5
Lesion type *		
Stellate		9
Architectural distortion		2
Calcification		2
Discrete mass		5
Non-specific density		3
Manufacturer		
GE Medical Systems	14	7
Philips Digital Mammography Sweden AB	8	5
Sectra Imtec AB	11	6
KODAK	0	1
Unknown	7	1

* A total of 21 lesions were found in the 20 cancer cases.

**Table 2 jpm-13-00888-t002:** Performance of models in detecting diagnostic errors on the testing dataset for radiologists from cohort A and cohort B.

	Normalized Dataset								
		Cohort A				Cohort B			
		Sensitivity	Specificity	Accuracy	AUC	Sensitivity	Specificity	Accuracy	AUC
False positive error	square	0.867	0.800	0.833	**0.882 (0.724–0.996)**	0.526	0.579	0.553	0.651 (0.457–0.817)
	RA	0.800	0.933	0.867	**0.920 (0.809–1.000)**	0.842	0.632	0.737	**0.787 (0.627–0.925)**
	whole	0.867	0.867	0.867	**0.911 (0.773–1.000)**	0.842	0.632	0.737	**0.780 (0.618–0.913)**
False negative error	square	0.625	0.375	0.500	0.523 (0.234–0.813)	0.800	0.600	0.700	**0.720 (0.450–0.930)**
	RA	0.875	0.500	0.688	0.695 (0.398–0.953)	0.800	0.700	0.750	**0.755 (0.490–0.960)**
	whole	0.500	0.375	0.438	0.375 (0.328–0.891)	0.800	0.600	0.700	**0.710 (0.450–0.930)**
False lesion location	square	0.571	0.429	0.500	0.469 (0.225–0.837)	0.556	0.222	0.389	0.556 (0.247–0.864)
	RA	0.714	0.571	0.643	**0.714 (0.388–0.959)**	0.778	0.667	0.722	**0.741 (0.457–0.963)**
	whole	0.714	0.429	0.571	0.367 (0.286–0.898)	0.778	0.444	0.611	0.654 (0.370–0.901)
	Non-normalized dataset								
False positive error	square	0.867	0.800	0.833	**0.933 (0.822–1.000)**	0.895	0.632	0.763	**0.774 (0.598–0.917)**
	RA	0.733	0.667	0.700	**0.813 (0.640–0.951)**	0.895	0.789	0.842	**0.831 (0.676–0.958)**
	whole	0.800	0.933	0.867	**0.933 (0.813–1.000)**	0.579	0.737	0.658	**0.702 (0.526–0.862)**
False negative error	square	0.750	0.625	0.688	0.812 (0.563–1.000)	0.800	0.700	0.750	0.750 (0.500–0.950)
	RA	0.875	0.500	0.688	**0.688 (0.359–0.922)**	0.900	0.500	0.700	**0.695 (0.450–0.910)**
	whole	0.875	0.750	0.813	0.867 (0.625–1.000)	0.700	0.600	0.650	0.745 (0.500–0.935)
False lesion location	square	0.429	0.286	0.357	0.347 (0.347–0.959)	0.556	0.556	0.556	0.611 (0.346–0.864)
	RA	0.571	0.429	0.500	0.367 (0.286–0.898)	0.556	0.222	0.389	0.494 (0.198–0.778)
	whole	0.714	0.286	0.500	0.418 (0.122–0.735)	0.556	0.333	0.444	0.531 (0.198–0.741)

The AUCs above 0.700 are in bold.

**Table 3 jpm-13-00888-t003:** Comparisons of AUCs between cohort A and cohort B using a 2000-time stratified bootstrap in detecting diagnostic errors.

Error Types	Normalization	ROIs	*p*-Values
False positives	Yes	square	0.042
		RA	0.144
		whole	0.195
	No	square	0.101
		RA	0.869
		whole	0.023
False negatives	Yes	square	0.323
		RA	0.746
		whole	0.653
	No	square	0.702
		RA	0.968
		whole	0.420
False location	Yes	square	0.812
		RA	0.892
		whole	0.920
	No	square	0.839
		RA	0.534
		whole	0.599

**Table 4 jpm-13-00888-t004:** The effect of ROI placement on error detection for readers from cohort A and cohort B.

Error Types	Radiologists	Normalization	ROIs	*p*-Values	95%CI
False positives	Cohort A	Yes	square vs. RA	0.493	−0.146~0.070
			RA vs. whole	0.903	−0.135~0.152
			square vs. whole	0.594	−0.135~0.077
		No	square vs. RA	0.206	−0.066~0.306
			RA vs. whole	0.191	−0.300~0.060
			square vs. whole	1.000	−0.150~0.150
	Cohort B	Yes	square vs. RA	0.175	−0.332~0.060
			RA vs. whole	0.940	−0.173~0.187
			square vs. whole	0.272	−0.359~0.101
		No	square vs. RA	0.474	−0.212~0.099
			RA vs. whole	0.264	−0.097~0.355
			square vs. whole	0.529	−0.152~0.296
False negatives	Cohort A	Yes	square vs. RA	0.341	−0.526~0.182
			RA vs. whole	0.779	−0.420~0.561
			square vs. whole	0.726	−0.669~0.465
		No	square vs. RA	0.445	−0.196~0.446
			RA vs. whole	0.225	−0.470~0.110
			square vs. whole	0.717	−0.350~0.241
	Cohort B	Yes	square vs. RA	0.818	−0.332~0.262
			RA vs. whole	0.743	−0.224~0.314
			square vs. whole	0.948	−0.292~0.312
		No	square vs. RA	0.606	−0.154~0.264
			RA vs. whole	0.553	−0.215~0.115
			square vs. whole	0.961	−0.196~0.206
False location	Cohort A	Yes	square vs. RA	0.295	−0.703~0.213
			RA vs. whole	0.119	−0.089~0.783
			square vs. whole	0.546	−0.229~0.433
		No	square vs. RA	0.905	−0.354~0.313
			RA vs. whole	0.671	−0.287~0.184
			square vs. whole	0.743	−0.499~0.356
	Cohort B	Yes	square vs. RA	0.357	−0.579~0.209
			RA vs. whole	0.601	−0.237~0.410
			square vs. whole	0.642	−0.516~0.318
		No	square vs. RA	0.465	−0.198~0.432
			RA vs. whole	0.862	−0.456~0.382
			square vs. whole	0.606	−0.225~0.385

**Table 5 jpm-13-00888-t005:** The effect of normalization on error detection for readers from cohort A and cohort B.

Error Types	Radiologists	ROIs	*p*-Values	95%CI
False positives	Cohort A	square	0.554	−0.220~0.118
		RA	0.267	−0.082~0.295
		whole	0.745	−0.156~0.112
	Cohort B	square	0.304	−0.358~0.112
		RA	0.676	−0.252~0.163
		whole	0.464	−0.130~0.285
False negatives	Cohort A	square	0.105	−0.639~0.060
		RA	0.967	−0.360~0.376
		whole	0.264	−0.667~0.183
	Cohort B	square	0.869	−0.387~0.327
		RA	0.636	−0.189~0.309
		whole	0.726	−0.231~0.161
False location	Cohort A	square	0.518	−0.248~0.493
		RA	0.119	−0.089~0.783
		whole	0.699	−0.309~0.207
	Cohort B	square	0.776	−0.438~0.327
		RA	0.213	−0.142~0.635
		whole	0.429	−0.182~0.429

## Data Availability

The data presented in this study are available upon request from the corresponding author. The data are not publicly available due to institutional regulations.

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
