# Peer review of "Using Radiomics-Based Machine Learning to Create Targeted Test Sets to Improve Specific Mammography Reader Cohort Performance: A Feasibility Study"

_jpm, 2023, doi:10.3390/jpm13060888_

Round 1

Reviewer 1 Report

This study explored the value of radiomics based machine learning to improve specific mammography reader cohort performance. The proposed method can be used to develop group-tailored mammographic educational strategies. This study is well-designed and innovative.

Author Response

We sincerely thank the reviewer for their valuable feedback on our paper. We are really grateful for their positive comment acknowledging that our study successfully explores the potential of radiomics-based machine learning in enhancing the performance of specific mammography reader cohorts.

Reviewer 2 Report

In this study, the authors developed a radiomics-based machine learning approach to reduce the errors in mammography reading by mapping diagnostic errors against global mammographic characteristics. The study is of interest in research community. I have minor comments to be implemented in the manuscript:

1. In Fig.3, the authors describe the classification approach. To obtain the final output, i.e., the classification score, they choose the maximum score of different output. Is it correct? Have you proven other 'ensemble approaches', e.g. soft voting? Could you better justify your choice?

2. Do you have any idea of how the normalization you applied affects the performance of the model? 

Author Response

  1. In Fig.3, the authors describe the classification approach. To obtain the final output, i.e., the classification score, they choose the maximum score of different output. Is it correct? Have you proven other 'ensemble approaches', e.g. soft voting? Could you better justify your choice?

Thank you for your valuable comment. We appreciate your input and have added sentences to justify our choice of using maximum operand.

To predict whether a mammographic case is challenging for a specific reader cohort, we fused the outputs, i.e., class probabilities from each of the image-wise classifiers built on each mammographic view. As suggested by the reviewer, various fusion methods can be explored. In studies focused on predicting diagnostic errors, such as ours, the maximum probability approach may be more appropriate, because it will ensure the sensitivity of the pipeline to detect challenging mammographic cases for education purposes. To be conservative, a case should be labeled ‘difficult’, regardless of the error status in the remaining views. On the contrary, the soft voting method sums up the class probabilities, takes the average, and the class with the highest value is considered the final label. However, using the soft voting method would average the likelihood of a case being labeled as 'difficult,' even if only one or two views present actual detection challenges. Additionally, we performed experiments using average scores and minimum scores, which resulted in lower AUC values compared to the maximum approach.

We also clarified the part in the manuscript in lines 201-208. We have also copied below.

“As shown in Figure 3, each pipeline returned multiple class probabilities indicating the likelihood of a given view being ‘difficult’. However, to obtain a case-wise prediction, we fused all outputs using the ‘maximum probability’ approach and only the one with the highest probability score was considered as the final output of the pipeline. This method was taken to ensure the pipeline’s sensitivity to detect challenging mammographic cases for educational purposes. Ideally, a case should be labeled as ‘difficult’ even if only one or two views present actual detection challenges, regardless of the error status in the remaining views.”

  1. Do you have any idea of how the normalization you applied affects the performance of the model? 

Based on the results presented in Table 5., we observed that feature normalization did not lead to a statistically significant improvement in the performance of the models. Additionally, in paragraph 5 of the Discussion (lines 451-463), we have elaborated on the underlying factors contributing to this observation. We hypothesize that in studies involving image perception (similar to ours), where radiologists analyze the original images instead of processed or normalized versions, the non-normalized features can truly reflect the radiologists’ clinical image processing procedure and thus perform well in diagnostic error prediction.

Reviewer 3 Report

Manuscript title: Using Radiomics-Based Machine Learning to Create Targeted Test Sets to

Improve Specific Mammography Reader Cohort Performance: A Feasibility Study

This study aims to reduce errors in mammography interpretation by using radiomics-based machine learning to map diagnostic errors against global mammographic characteristics. The approach successfully predicted false positive and false negative errors of two cohorts of radiologists, but not consistently predicted location errors. 

In the abstract author mentioned “Feature normalization did not offer additional benefits. ” Please rewrite the sentence in context.

One of the major concerns of the study is the relatively small sample size which may limit the generalizability of the results to larger populations.

Author should use external validation dataset possibly from TCGA/other data to compare and validate model.

The machine learning part and the selection criteria of features are not described properly. Please add a flowchart also.

How the ROI was drawn as in figure2.

Author Response

  1. In the abstract author mentioned “Feature normalization did not offer additional benefits.” Please rewrite the sentence in context.

Thank you very much for raising this point. We have revised the sentence in the abstract, see lines 19-20:

“The performance of the models did not show significant improvement after feature normalization, despite the mammograms being produced by different vendors.”

  1. One of the major concerns of the study is the relatively small sample size which may limit the generalizability of the results to larger populations. Author should use external validation dataset possibly from TCGA/other data to compare and validate model.

Thank you very much for your comments. We acknowledge the concern regarding the relatively small sample size and its potential impact on the generalizability of our results to larger populations. In response, we have further discussed the limitations of our study in the Discussion section and outlined potential avenues for future research. Please refer to lines 359-380.

This is a proof-of-concept study to show the feasibility for the future studies, and certainly larger future validation studies are required.  We agree that external validation using publicly available datasets would be valuable for model validation. However, for our specific study, which focuses on predicting reader diagnostic errors, current publicly available datasets containing mammographic examinations with cancer/normal ground truths are not suitable. These datasets are primarily designed to develop AI models for diagnosis, while our objective is to predict the likelihood of diagnostic errors made by radiologists. Additionally, acquiring our dataset was resource-intensive, with each radiologist spending approximately 2 hours on the test set. In total, the data collection process required 72 hours from participating radiologists.

Despite the sample size limitation, our study has notable strengths. We have radiological assessments from 36 radiologists with varying levels of experience, and our mammographic dataset is enriched with cancer cases representing various lesion subtypes commonly encountered in breast screening. This ensures representative observer performance, which is crucial in diagnostic error analysis studies like ours.

Moreover, as a proof-of-concept study, we hope our research stimulates reflection on the current one-size-fits-all radiology training system, encouraging a collective effort to revolutionize radiological education. Our study proposes novel ideas for integrating state-of-the-art computer vision technology into radiology and demonstrates the feasibility of using machine learning to predict the difficulty level of radiological cases. We encourage researchers to embrace this concept, leveraging the potential of AI to improve radiology performance, and we urge the community to create more comprehensive and extensive reader performance datasets.

Furthermore, our study highlights the importance of global radiomic features in predicting diagnostic errors. These features have previously shown significance in shaping radiologists' initial impressions and guiding their subsequent evaluations. We have demonstrated their potential in cohort-specific diagnostic error prediction. Therefore, we encourage researchers to explore these features, alongside localized features, to uncover their potential in various tasks such as diagnostic error prediction, breast cancer detection, and breast cancer prediction.

  1. The machine learning part and the selection criteria of features are not described properly. Please add a flowchart also.

Thank you for your comment. To provide a clearer description of the machine learning and feature selection process, we have revised the manuscript. Please refer to lines 191-255 for detailed information.  

Additionally, we have included a new flowchart at lines 238-255.

  1. How the ROI was drawn as in figure2.

Thank you for your comment. We used an open-source framework for mammography image analysis to draw the ROI. Please, see lines 146-147:

“This was done using the open framework for mammography images analysis in Matlab_R2022b (Mathworks, USA) [17-19]”.

Round 2

Reviewer 3 Report

Comments addressed to satisfactory!